# WasmTree: Web Assembly for the Semantic Web

Julian Bruyat[1], Pierre-Antoine Champin[1,2], Lionel Médini[1], and Frédérique Laforest[1]

[1] Université de Lyon, INSALyon, UCBL, LIRIS CNRS UMR 5205, France
[2] W3C / ERCIM, France

**Abstract** Today, Javascript runtimes intend to process data both at server and client levels. In this paper, we study how Rust and Web Assembly can contribute to implement efficient Semantic Web libraries for the Javascript ecosystem. We propose WasmTree, a new implementation of the RDF.JS `Store` and `Dataset` interfaces in which the processing is carefully split between the Web Assembly and Javascript layers. Experiments show that our best setup outperforms state-of-the-art implementations for fine-grained data access and SPARQL queries.

**Keywords:** Semantic Web · Web Assembly · RDF.JS · Indexing

## 1 Introduction

Nowadays a large number of RDF libraries help application developers take advantage of Linked Data and the Semantic Web. On the server side, when high performance is needed, it is usual to use compiled languages, such as C or Rust. But until recently, in browsers, only libraries written in Javascript (JS) could be used, such as N3.js [16] or Graphy [13]. With the development of Web Assembly [14] (WASM), browsers and other JS runtimes are able to run compiled and highly optimized libraries with near-native performance.

Our aim is to build an efficient RDF library for JS runtimes. We propose WasmTree[34], a fast in-memory implementation of the RDF.JS API in which the processing is carefully split between the WASM and JS layers. Section 2 presents the state of the art of RDF libraries for JS environments. Section 3 presents several attempts to implement efficient Semantic Web libraries using Rust and WASM, the last being WasmTree. Section 4 evaluates these approaches in comparison with other libraries. Finally, we conclude with open issues.

## 2 State of the art

**RDF Dataset:** RDF [9] is the core data model of the Semantic Web. The building block of RDF data is a *triple* composed of a subject (an IRI or a blank node), a predicate (an IRI) and an object (an IRI, blank node, or literal). A

---

[3] https://www.npmjs.com/package/@bruju/wasm-tree
[4] https://github.com/BruJu/WasmTreeDataset/tree/master/wasm-tree-frontend

set of triples is an *RDF graph*. As graphs are rarely used in isolation, the RDF recommendation defines the notion of *RDF dataset*, composed of one default graph and zero or more named graphs. A dataset can therefore be seen as a set of *quads*, composed of a subject, a predicate, an object, and a graph name (the latter being an IRI, a blank node, or a special marker for the default graph).

**Indexing:** The most straightforward data structure for representing an RDF graph or dataset is to store triples or quads with their constituting terms directly in the data structure, like in Graphy [13]. Another approach is to store, on one side, a mapping between RDF terms (IRIs, literals, blank nodes) and short identifiers (usually integers), and on the other side, a set of triples or quads where the constituting terms are referred to by their identifiers. This is the approach chosen by HDT [10]. This approach is more memory-efficient and makes it easier to use sorted data structures. On the other hand, it requires an additional step to map identifiers and terms when quads are ingested or retrieved.

**RDF in JS runtime environments:** In order to foster interoperability in Semantic Web developments on JS runtimes (browsers, Node.js[5], Deno[6]...), the RDF.JS W3C Community Group[7] was formed in 2013. It has proposed three specifications [3,4,5] defining APIs for RDF building bocks, datasets and streams. In the two latter, a `Dataset` interface and a `Store` interface are defined, a store being a dataset usable in an asynchronous fashion. The most prominent implementations of these APIs are Graphy for `Dataset` which uses 1 indexing tree (by Graph then Subject then Predicate then Object, or GSPO) and N3.js for `Store` which stores the data in 3 different indexes (SPO, POS and OSP) for each graph.

Another notable project using the RDF.JS interfaces is Comunica [15], a highly modular framework allowing, in particular, to execute SPARQL [11] queries against any `Store` implementation.

**Web Assembly:** Most JS runtimes nowadays make use of Just In Time (JIT) compilation, which largely increases the performance of JS code. However, since 2015, another approach has been explored, which lead to the standardization of Web Assembly (WASM) in 2019. WASM is a low-level binary language which, alongside JS, is executable in most JS runtimes. WASM files are much smaller than equivalent JS files, and are compiled *ahead of time*, saving time at execution and opening the way to more aggressive optimizations. WASM is executed in a virtual machine in order to be portable, but at the same time is very close to machine code, in order to achieve near-native performance.

For security and portability reasons, WASM code can only work on a *linear memory*. More precisely, this linear memory is allocated by JS code as an array

---

[5] https://nodejs.org/

[6] https://deno.land/

[7] https://www.w3.org/community/rdfjs/

of bytes, and provided to the WASM code. Communication between JS code and WASM code is only possible through function calls returning integers or floating point numbers and modifications in this array. At the time of writing, a dozen languages can be compiled to WASM[8], Rust being one of them.

**Rust:** [12] is a programming language created by Graydon Hoare, first released in July 2010 and supported until 2020 by Mozilla. Rust emphasizes performance, reliability and productivity. Performance is ensured by the fact that everything in Rust is explicit; for example, Rust has no exception mechanism nor garbage collector. Reliability is ensured by an original ownership model, guaranteeing memory-safety and thread-safety. Yet, productivity is made possible by powerful abstractions, carefully designed to have minimal or zero cost at runtime.

A few RDF libraries exist in Rust. Sophia [8] provides generic interfaces for graphs and datasets, aiming to play the same role as RDF.JS in the JS ecosystem. Oxigraph[9] provides a persistent dataset storage and SPARQL query engine.

Rust was one of the first languages that could be compiled to WASM. What sets Rust apart from other WASM-enabled languages is the wasm-bindgen tool. We have described above how the WASM virtual machine can only work on and communicate through its linear memory, which is seen by JS code as an array of numbers. Communicating other data types to JS code requires an additional layer of JS "glue code" to re-interpret the content of the linear memory into the corresponding abstraction. Such glue code is automatically generated by wasm-bindgen, requiring only that the Rust code be annotated using a set of dedicated keywords. To the best of our knowledge, Rust and C++ (with Emscripten) are the only languages equipped with such tools.

## 3   Towards an efficient implementation of RDF.JS

We have developed a fast in-memory implementation of the RDF.JS `Dataset` interface usable in any JS runtime. To ensure efficiency, we have developed a new dataset structure in the Rust language. Based on the WASM principles, it is compiled and exported to JS runtimes using the wasm-bindgen tool, so that JS runtime developers can use it from their JS code. We first present two approaches that are more straightforward ways to implement a RDF.JS library using WASM, but underperform. Then based on these observations, we present WasmTree.

### 3.1   General view of our approach

Our `Dataset` structure implementation is separated into two parts:

- A bidirectional *term mapping* to store the correspondence between terms and their integer identifiers.

---

[8] https://webassembly.org/getting-started/developers-guide/
[9] https://github.com/oxigraph/oxigraph

– A *forest* of B-Trees [2]. Each B-Tree sorts quads of identifiers in a given order e.g. SPOG, GPOS. Details are provided hereafter.

The term mapping is used to convert quads of terms into quads of identifiers and conversely. To provide constant lookup time, this mapping is implemented as a hashmap to map terms to identifiers, and a vector to map identifiers to terms (identifiers are assigned in sequence). Details about how this mapping is implemented and used are provided in the following subsections.

The quads of identifiers are stored in several B-Trees. The reasons are the followings:

– If we classify search patterns according to which positions (Subject, Predicate, Object or Graph name) are fixed (as opposed to a wildcard), there are 15 classes of patterns (the "all-wildcards" pattern is not interesting). For each class of patterns, we need a B-Tree sorted by the fixed positions, so that all matching quads can be efficiently located and iterated over.
– While up to 16 different sort orders are possible (SPOG, OSPG, OPSG . . . ), 6 sort orders are sufficient to optimally answer any pattern. Indeed, when a pattern is queried, any sort order starting with the fixed positions of the pattern is appropriate; for example if the subject (S) and the object (O) are fixed (which we will refer to as a S?O? in this paper, using ? as a wildcard), any of the following trees could be used: **SO**PG, **SO**GP, **OS**PG and **OS**GP. Note that this is possible because the order of the resulting quads can not be specified using the RDFS.JS APIs.
– As all classes of patterns will not necessarily be used in practice, we initialize our structure with only one B-Tree. The other five B-Trees are built lazily when a pattern is queried for which the available trees are not suited. This enables to speed up the initial dataset load. For example a tree that starts with SO will only be required when the pattern S?O? is queried.
– B-Trees are preferred over binary trees to benefit from the principle of locality of reference[10]: by using B-Trees, up to 2B-1 quads can be stored in the same node, which is more cache friendly. This choice is also reinforced by the fact that the Rust standard library offers functions to directly extract a range from a B-Tree, which helps code maintainability.

This B-Tree structure is common among all our implementations and implemented in Rust. On the other hand, the management of the term mapping, and more generally the way to retrieve the stored quads from WASM memory to the JS code, comes in three different propositions:

– The naive TreeDataset approach is a full-Rust implementation of the dataset, which is compiled in WASM and exposes individual quads to JS.
– The all-at-once TreeDataset approach also uses a full-Rust implementation of the dataset compiled in WASM, but reduces the number of exchanges between WASM and JS code by transmitting all quads as one single string serialized using the N-Quads format [7].

---

[10] https://en.wikipedia.org/wiki/Locality_of_reference

– The WasmTree approach implements only the B-Trees in Rust, while the term mapping is managed by JS code.

## 3.2   Naive TreeDataset approach

In this first approach, we have implemented both the B-Trees and the term mapping in Rust, in a structure called TreeDataset. More precisely, TreeDataset complies with the Sophia API [8]. The Sophia and the RDF.JS APIs are quite similar. In particular, both provide methods to retrieve a subset of quads matching a given pattern. The main difference is the return value of pattern matching methods: Sophia returns a stream of quads that can be lazily evaluated, while the `Dataset` interface returns a new dataset containing all the match quads.

To hash a term, we use the default Rust hasher implementation. Every part of the term is hashed successively and then the hash code is produced with the SipHash algorithm. For example, if the term is an IRI, the Rust hasher is used on that IRI. If the term is a non language literal, the hasher is used on both the IRI and the datatype.

By writing an adapter annotated with wasm-bindgen, the dataset structure is compiled in WASM. Thanks to the generated JS glue code, this implementation can be used in a JS code like any other RDF.JS `Dataset` implementation.

Figure 1 shows the process to iterate on all quads. An iterator object is first constructed from the list of all quads stored in the output dataset. This iterator is then exported to the JS user. When the user code requests the next quad, the returned quad is produced by Rust code.

As shown later in Section 4, naively adapting through WASM a Rust API to a JS API, however similar, leads to bad performance. Indeed, there is a cost for every individual exchange between the JS code and the WASM code, meaning it is better if data is sent in bulk[11].

Another drawback of this approach is related to memory leaks. Unlike JS, WASM does not have a garbage collector. While the WASM memory is implicitly allocated by classes generated by the JS glue code when the constructor is used, the user has to explicitly free it using the `free` method. This means that the whole datastructure representing the dataset in WASM needs to be explicitly deallocated, which is uncommon for JS developers and is not part of the RDF.JS specification. Recent versions of JS introduce the FinalizationRegistry, which can be used as a destructor: this is used by recent versions of wasm-bindgen. So when the JS object that owns a part of the WASM memory is destroyed, the memory is freed. In our implementation, we prefer to rely as little as possible on allocated objects to ensure compatibility with older versions of JS runtimes, but also provide a library that conforms to JS developers current idioms.

To solve these performance issues, the next sections present two other approaches aiming at mutualizing exchange costs and reducing the number of WASM allocated objects.

---

[11] As a side experiment, we tried to send 1,000,000 integers from WASM to JS: sending in bulk the whole array was 8 times faster than sending the integers one by one.

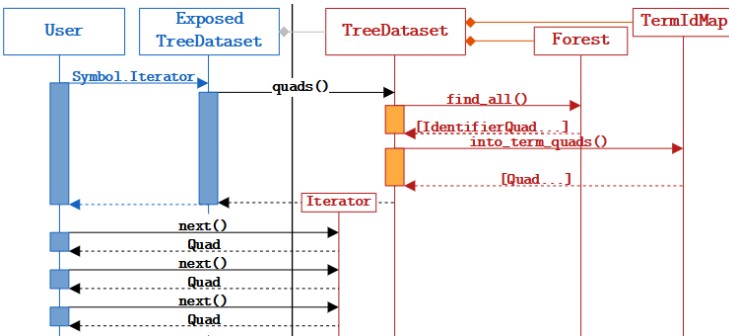

**Fig. 1.** Process to retrieve quads from a TreeDataset - naive TreeDataset approach. The blue part in is JS, while the red part is a WASM built from Rust

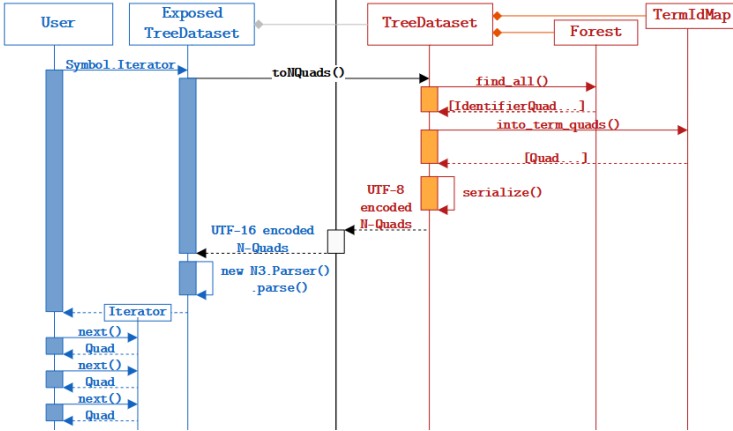

**Fig. 2.** Process to retrieve quads from a TreeDataset - All-at-once TreeDataset approach. The blue part in is JS, while the red part is a WASM built from Rust

### 3.3 All-at-once TreeDataset approach

We tried a different strategy for iterating over quads: rather than retrieving a single quad from the WASM dataset at each iteration, we export all quads at once, using a textual serialization format.

Figure 2 presents the different steps of this approach: we first serialize all the quads in the N-Quads format [7] into a big string. JS retrieves this string by calling the `toNQuads` function, which first takes care of transcoding it from UTF-16 to UTF-8, then uses the N-Quads parser of N3.js to get a list of quads.

Unfortunately, our experiments described in Section 4.2 show that this approach is slower than the previous one. This is mainly due to the high cost of interpreting the N-Quads formatted string received by the JS code, even though N-Quads are a very simple and straightforward format.

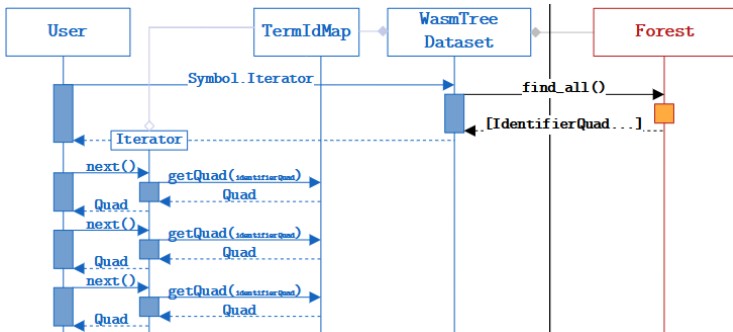

**Fig. 3.** Process to retrieve quads from a TreeDataset - WasmTree approach. The blue part in is JS, while the red part is a WASM built from Rust

### 3.4   WasmTree: A hybrid implementation

Our last implementation, which is the main contribution of this work, consists in splitting wisely the work between WASM code and JS code. Previous approaches fail because of the cost of exchanging and reinterpreting complex data (in our case objects and strings), as they are not trivially copyable[12] from the WASM memory to the JS memory. So we decided to do computation-intensive integer manipulation (i.e. the construction, storage and manipulation of the B-Trees structure) in WASM, and to handle strings (i.e. the mapping between identifiers and terms) in JS. Thus, only integers need to be exchanged across the WASM-JS boundary. More precisely, the JS part implements the RDF.JS `Dataset` interface with three main components:

- The term mapping, implemented with an array and a standard JS dictionary; as the array only accepts simple strings as keys, we use the notion of concise term[13] to unambiguously encode RDF terms into strings.
- A handle to the WASM structure that contains the forest of B-Trees.
- An identifier list representing the quads in the dataset. Elements of this list are read four at a time (subject, predicate, object, graph name). The role of this list is to serve as a cache to limit exchanges with the WASM memory.

When a new WasmTree instance is created by the user, it is created with no forest and no identifier list. Iterating over the quads of the dataset is performed using the identifier list, as illustrated in Figure 3. If the list is absent, it is first recomputed from the forest. If there is no forest, an empty identifier list is used instead. When methods modify the content of the dataset (`add`, `addAll`,

---

[12] By "trivially copyable", like in C++, we mean that to copy the data, the underlying bytes can be copied from one location to another. An array of 32 bits integers is trivially copyable from WASM to JS with the Int32Array object type. A string is not trivially copyable because of the different encodings.

[13] https://graphy.link/concise

`delete`. . . ), the terms received as parameters are converted to identifiers, the changes are applied to the forest, and the identifier list is deleted as its content no longer reflects the content of the forest.

The `match` method takes a quad pattern and builds a new dataset containing only the matching quads. First, the terms in the pattern are converted to identifiers, and the forest is requested for matching quads (it is more efficient than browsing the identifier list). This produces a new identifier list, which is used to build a new dataset containing that list, but no forest. The identifier list returned by the forest in WASM is a plain `Uint32Array` Javascript object, meaning that the garbage collector is able to manage it. It is also cheap to produce, as it only requires to copy a segment of the WASM linear memory. The mapping of terms is not duplicated, but shared with the original dataset.

The benefit of this design is that, once an identifier list has been retrieved, and as long as that dataset is not modified, no exchange is required with WASM. Furthermore, while the `match` method is required by the RDF.JS API to return a full-fledged dataset, in most cases, this dataset will be iterated over and dismissed, therefore spending time indexing its quads into B-Trees would be useless. However, the indexing (forest reconstruction) will happen lazily if the dataset was to be modified or queried again with `match`.

Our `Dataset` interface also provides a `free` method which is not part of the RDF.JS API. It removes the forest and the identifier list, and frees the WASM memory segment allocated to the former. From the the end-user's point-of-view, it has the effect of emptying the dataset.

By using a similar split between WASM and JS code, we also developed an implementation of the RDF.JS `Store` interface. The main differences between `Dataset` and `Store` are that `Store` methods are asynchronous and the `match` method in `Store` returns a stream of quads instead of a new dataset. Our implementation is based on an asynchronous call on a function which starts with retrieving the identifier list corresponding to the pattern, and then rebuilds a quad by using the identifier list and the JS dictionary every time a `data` message is emitted.

## 4   Evaluation

In this section, we evaluate the performance of WasmTree in three different situations: first, we study the amounts of time and memory required to initialize a dataset from an N-Quad file, second we evaluate its performance on a simple task of pattern matching, third we evaluate it in the case of a SPARQL query.

All these evaluations have been performed using the following setup. Compilation tasks have been performed using Rust Compiler 1.43.0, and wasm-bindgen 0.2.63. Benchmarks have been coded and run on the NodeJS 10.19.0 platform. API tests have been performed on a PC equipped with an Intel(R) Core(TM) i5-1035G1 processor, 16GB of DDR4 RAM, and the Ubuntu 20.04 LTS OS. SPARQL tests have been performed on a virtual machine with 4 VCPUs (based on 2600 MHz Intel Xeon Skylake processors) and 8GB of RAM.

|             | # of indexes | Time (s) | Memory (kB) |
|-------------|:------------:|:--------:|:-----------:|
| *TreeDataset* | *1* | 10.825 | ***30,982*** |
| *TreeDataset* | *6* | 12.976 | ***30,994*** |
| *WasmTree* | *1* | **5.343** | 411,093 |
| *WasmTree* | *6* | 7.457 | 407,511 |
| *Graphy* | *1* | **5.653** | 595,890 |
| *N3.js* | *3* | 6.985 | 1,286,967 |

(a) Time and memory used at initialization.

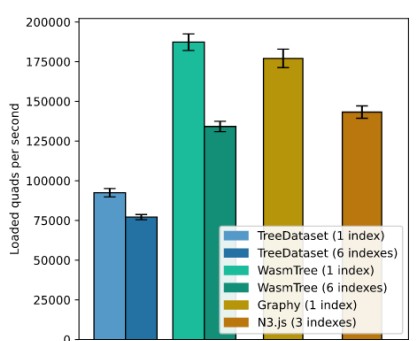

(b) Initialization speed. Higher is better

**Fig. 4.** Time and memory used to initialize a 1M quad dataset in various RDF.JS implementations.

The two first experiments (dataset initialization and simple pattern matching) were conducted under the following conditions: the datasets we used are extracts of various sizes (up to 1 million quads) from the *Person data* dataset of the 2016-10 dump[14] of DBpedia [1] in N-Triples format, considered as an RDF dataset containing one default graph. All measures were performed 50 times and the average was computed. Both experiments were performed using the RDF.JS API and the N3.js parser to read the dataset. A step by step guide to reproduce the experiments can be found at https://github.com/BruJu/wasm_rdf_benchmark/tree/eswc2021.

### 4.1   Evaluation of dataset initialization

We herein evaluate the cost, in time and memory, of initializing a dataset from an N-Quads file containing 1 million quads, in different implementations. By initializing, we mean here all the operations required to make the dataset available to the JS application: reading and parsing the data, and populating data structures. For the TreeDataset approaches and WasmTree, we tested a lazy configuration, where only one of the six B-Trees is constructed (default), and a greedy configuration immediately building all six B-Trees.

**Time:** Figure 4b shows initialization speed for several RDF.JS `Dataset` implementations. We also compare N3.js using its synchronous `addQuad` function. With only one index, WasmTree is 1.30 times faster than N3.js, and is 1.06 times faster than Graphy, which doesn't resort to a dictionary and directly stores quads in its hashtree. When building greedily its 6 indexes, WasmTree is 1.32 times slower than Graphy, and 1.06 times slower than N3.js.

---

[14] https://wiki.dbpedia.org/develop/datasets/downloads-2016-10

The table in Figure 4a shows that the difference between filling one and six indexes is the same (2.1 seconds) for our two implementations. This is not surprising as they use the same data structures for indexes. This corroborates our hypothesis that the poor performance of TreeDataset are related to storing strings in the term mapping in the WASM memory. The lost time is more impacting in terms of initialization speed for WasmTree as a 2.1s difference represents a higher ratio of the total time. In real world applications, the lazy indexing strategy will ensure the users benefit from the fastest initialization time, amortizing the cost of creating the missing indexes over the use of the dataset.

**Memory:** The table in Figure 4a also exhibits an estimation of the memory used to store the different benchmarked RDF.JS implementations. These estimations are based on the peak virtual memory usage, as provided in Linux by the `VmPeak` field of the file `/proc/self/status`. This introduces a bias because JS may allocate memory before the garbage collector has freed memory blocks that are not used anymore. Actually, we see that this method is not precise enough to measure the difference between one and six indexes within the same implementation, therefore the most relevant information from these measures is the order of magnitude.

We can see that implementations relying on WASM are more compact than pure JS implementations. TreeDataset, used by the naive approach and the all-at-once approach, entirely based on Rust data structures stored in the WASM memory, is by orders of magnitude the most memory-efficient. However, its initialization time and its poor query performance (as shown in Section 4.2) make it non satisfying. The hybrid approach (WasmTree) still consumes less memory than the best pure JS implementation in our study (Graphy).

The difference between pure Rust implementations (TreeDataset), those using fewer JS structure (WasmTree, Graphy) and those using many JS structures (N3.js), demonstrates that Rust structures compiled into WASM are much more compact than comparable JS structures. Additionally, we already remarked that the memory used by WASM B-Trees is negligible compared to the term mapping and the noise induced by the garbage collector. This motivates our choice to store several indexes in WASM memory in order to improve speed.

### 4.2   Evaluation of simple pattern matching

We now proceed to evaluate the performance of extracting quads matching a given pattern. This corresponds to simple queries such as "find all quads about person X" or "find all resources of type Person".

For each measure, we load a dataset and call the method `match` twice with the given pattern, measuring the time taken by the second call. The first method call ensures that lazily built indexes and caches are computed. The cost of these operations is considered irrelevant here as 1) it will be amortized on multiple calls, and 2) different implementations will trigger them at different times or for different queries, making comparisons less meaningful. We then measure the

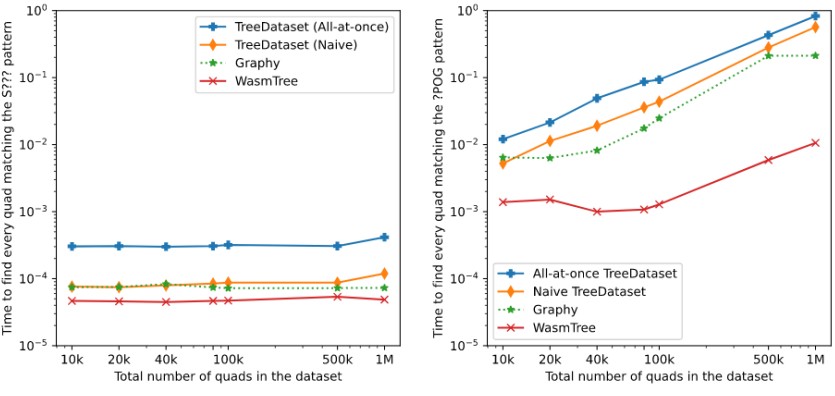

(a) S??? pattern, constant-size output    (b) ?POG pattern, linear-size output

**Fig. 5.** Time (in seconds) to iterate over quads matching a given pattern.

time it takes to iterate over all the quads of the `Dataset` returned by the second call to `match`.

**Performance with different patterns:** The first set of measures we ran is for querying the dataset for quads having a given subject. We chose that subject in such a way that the number of matching quads was the same (seven) regardless of the dataset size. TreeDataset and WasmTree use their SPOG index to answer this query. The unique index used by Graphy is also optimal in this case (GSPO, with only one graph in the dataset). The results are shown in Figure 5a.

TreeDataset is slower than Graphy, a pure JS implementation. This can be explained by the fact that the time saved by using WASM is lost in the translation process with JS. The all-at-once approach is the worst, being twice as slow as the naive approach. The reason for these poor performance will be explored later with Figure 6a. On the other hand, WasmTree reduces the number of exchanges (one array describing all the quads instead of individual quads) and the complexity of the exchanged data (integers rather than strings). This reaches the best trade-off, where the manipulations of numbers are most efficiently handled by WASM code, while string manipulations are left to JS.

We see that the size of the input dataset has only a minimal impact on all implementations. This shows that the choice of B-Trees, that have a $\mathcal{O}(\log n)$ complexity for searching quads, is a viable alternative to hashmaps with $\mathcal{O}(1)$ complexity (as used by Graphy).

Figure 5b shows the result of a second set of measures, where we are now looking for every resource of type Person in the default graph. For every dataset size, the retrieved quads represent about 13% of the total dataset size. In other words, the output size in this benchmark is linear with the dataset size. For this query pattern (?POG), Graphy's GSPO index is not theoretically optimal,

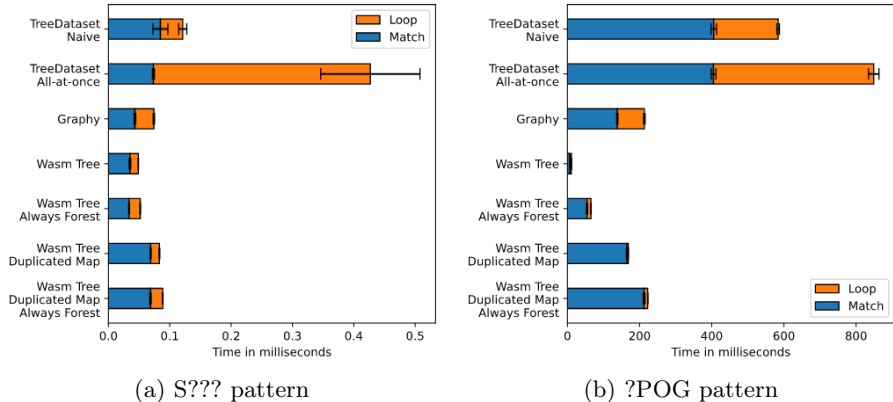

(a) S??? pattern          (b) ?POG pattern

**Fig. 6.** Workload distribution when matching over a certain pattern.

which would make the comparison unfair, as our implementation always uses an optimal index. However, in this particular dataset, all the subjects are of type Person, so they all match the pattern. Graphy is therefore not sanctioned for having to loop on all the subjects.

We can see that the order remains the same: WasmTree is still the most efficient solution, followed by Graphy and then TreeDataset. The gap between different options is bigger, and increases with the size of the output: instead of being 1.5 times faster than Graphy as in Figure 5a, WasmTree is now 4 to 35 times faster than Graphy, scaling better as the output size grows.

We also benchmarked our implementation of the `Store` interface using Wasm-Tree and compared it to N3.js in similar conditions. We found that WasmTree also outperforms N3.js. For example, when looking for every person in the dataset, WasmTree is on par with N3.js for small datasets, but becomes increasingly faster (up to 13.671 times) when the input dataset grows. Using the `Store` interface of WasmTree makes the iteration over all quads 2.73 to 8.22 times slower than using its `Dataset` interface; it means that in practice, consuming an asynchronous stream of quads is more time consuming than populating an intermediate dataset.

**Workload distribution:** In Figure 6, we focus on the `Dataset` interface, to study how the overall time measured previously is distributed between finding the matching triples (match), and iterating over them (loop)[15]. While Figure 6a shows measures for the same query as Figure 5a, using a S??? pattern, Figure 6b shows measures for the same query as Figure 5b, using a ?POG pattern. In both figures, we use the biggest version of our test dataset (1M quads).

---

[15] This distinction is not relevant for the `Store` interface, where the iteration is performed in an asynchronous fashion.

In TreeDataset, the all-at-once approach is very inefficient: the looping part is up to 11 times slower than the naive approach, especially when the returned dataset is small (Figure 6a). Profiling the code reveals that 22% of the time spent is used by the garbage collector. Compared to the naive approach, the all-at-once approach aimed at reducing the number of exchanges, but that is at the expense of a more complex processing to retrieve all quads. Indeed the implementation has to change the encoding of the string from UTF-8 to UTF-16, and then parse the string to build the quads.

TreeDataset is always slower to loop on all quads than Graphy. The total time for the latter is even lower than the match time for TreeDataset. The match operation has limited exchange with WASM: it consists in only one function call. We can deduce that the cost of interpretation of the WASM code and our code are not efficient enough to beat Graphy, both for small and big results. It can be explained because Graphy does not build a whole new dataset when the `match` method is called: the returned dataset shares data structures with the source dataset, unlike TreeDataset that copies the quads and creates a new independent dataset.

WasmTree, on the other hand, is faster than all the other approaches, especially during the loop, and when the output dataset is big (Figure 6b). This confirms that the computation intensive part of the processing is mostly handled by WASM code (in the `match` method), making quick and easy the remaining task of the JS code handling the loop. Thanks to the compactness of the B-Trees stored in the WASM memory, we can afford to store indexes for all possible patterns, while Graphy is bound, with a unique index, to be less efficient for some patterns.

**Ablation study:** WasmTree uses two different strategies to be fast. The first one consists in caching the list of identifiers in an array, as described at the end of Section 3.4. The second one consists in sharing the term mapping between a dataset and the (sub-)datasets returned by the `match` method. In Figures 6a and 6b, we studied the impact of these optimizations by removing the first one ("Always Forest"), removing the second one ("Duplicated Map") and removing both.

The sharing of term mappings is what saves most time, dividing by 2 the total time for the S??? benchmark. Producing an independent dataset when the `match` method is called is expensive, as it requires to produce and populate a new term mapping, and this operation has to be done in JS. It results in similar to slightly worse performance as compared to Graphy. When producing a dataset that shares the term mapping, WasmTree is 1.4 to 19 times faster than Graphy, which shares its index and has no mapping.

The caching of the identifier list has a bigger impact when the output dataset is large (?POG benchmark). Indeed, when there are few elements, the identifier list memory representation in JS and the B-Tree one in WASM are close. In the S??? benchmark, as there are only 7 quads to return, both representations are an array of 28 integers. In the default WasmTree implementation, the identifier list is

already in the JS environment, so it does not call a WASM function to retrieve it. The "WasmTree Always Forest" implementation retrieves the identifier list from the WASM environment. When there are less than 2B elements, it consists in simply copying the root element of the B-Tree. This is the only difference between the two, and explains the small measured difference in time, as shown in figure 6a.

In every benchmark but memory consumption, WasmTree, the implementation that splits works between WASM and JS is better than TreeDataset: having faster initialization time, faster quad retrieval and being less error prone, as users do not have to care about freeing memory except for the datasets. Moreover, the identifier list permits users to forget about freeing the datasets produced by `match`, as they do not actually allocate WASM memory.

### 4.3   Evaluation of SPARQL queries

To get a broader insight on performance, our third set of experiments evaluates performance on SPARQL[11] queries. SPARQL allows to express complex queries, contrarily to the simple query patterns studied in the previous section.

We used the Berlin SPARQL Benchmark [6] (BSBM), a benchmarking framework for SPARQL. It can generate synthetic datasets about products, vendors and consumers, and realistic queries over these datasets. It provides a driver that can submit a set of queries (a *query mix*) to a SPARQL endpoint and measure its performance. We are interested here in information retrieval: the driver sends a query mix made of 25 queries, generated from 12 query templates from the "Explore" use-case of BSBM. For example, in the first template "Find the name of all products having [two specific features]", the driver randomly chooses the two features for each query.

As described in Section 2, Comunica can be used to setup a SPARQL endpoint above any implementation of the RDF.JS `Store` interface. We use Comunica v1.13.1 to expose a SPARQL endpoint above our RDF.JS implementation. In this section, we will compare the default configuration (using N3.js) with a Comunica SPARQL endpoint backed by WasmTree.

Oxigraph, which we also described in Section 2 is a Rust library implementing a dataset and a SPARQL engine. It can be compiled to WASM, providing its own API which only partially complies with RDF.JS, but exposes SPARQL functionalities. It was therefore relatively straightforward to use it to build a Node.js-based SPARQL endpoint, where the whole query processing is done in WASM.

**Performance** Table 1 shows that, when it comes to executing SPARQL queries, replacing N3.js with WasmTree as Comunica's store allows to execute almost 7 times more queries in the same time. WASM increases performance even when used only for a part (finding quads) of a more complex operation (executing a query plan and computing joints). Oxigraph is itself 20 times faster than Comunica with WasmTree. The difference can be explain both by the efficiency

**Table 1.** Performance of three SPARQL endpoints, with a dataset of 2000 products (725305 quads). Measured from 100 executions after a warmup of 20 executions.

| | *Query mixes per hour* | Acceleration compared to | | |
|---|---|---|---|---|
| | | Oxigraph | Comunica +WasmTree | Comunica +N3.js |
| *Oxigraph* | 8877.66 | | ×19.90 | ×135.25 |
| *Comunica +WasmTree* | 446.22 | ×0.05 | | ×6.80 |
| *Comunica +N3.js* | 65.64 | ×0.007 | ×0.15 | |

of WASM for complex processing paying off compared to the cost of transferring the results, but also the fact that Oxigraph is not limited by the RDF.JS API, which enables it to use better query plans.

**Memory limitations** We initially tried to run the BSBM driver with a scale factor of 10,000 (corresponding to a dataset of 4M quads), but we ran into memory shortage. For Comunica-based approaches, this was solved by increasing the memory allocated to JS. However, for Oxigraph, the limitation was due to WASM itself, which uses 32bit addresses, and is therefore structurally limited to handling 4GB of linear memory. This limit is easily reached with big RDF datasets. The strict limitation is less impacting on WasmTree, because it only uses WASM memory for BTrees.

## 5   Conclusion and perspectives

In this work, we studied how we can use Rust and WASM to build efficient RDF libraries for JS runtimes. Our experiments show that using WASM naively does not improve performance. In an effort to reduce the costs related to exchanging information between WASM and JS, we have proposed WasmTree, a hybrid implementation providing an efficient distribution of tasks between the two languages. On simple pattern matching queries, it outperforms two state-of-the-art JS libraries: Graphy and N3.js.

In the context of SPARQL query evaluation, the fastest approach by far is the one relying entirely on WASM (Oxigraph). However, the WASM intrinsic memory limitations hinder the scalability of this approach. WasmTree, on the other hand, improves the performance of the Comunica framework while being much less affected by this scalability issue.

However, the forest structure, which is the only part implemented in Rust for WasmTree, has no access to the actual terms in the graph. Thus, semantic processing such as inferences, which could benefit from the performance gain of WASM, can not directly be implemented on the WASM side. To tackle this problem, we should refine the hybridization between the two sides of our implementation. some clues are the following: we could fix once and for all the

identifier of semantically loaded IRIs (such as `rdfs:subClassOf`), or enable the term mapping to share some specific identifiers with the index, depending on the inference ruleset. These ideas will be for further work.

**Acknowledgment** This work was partly supported by the Fédération Informatique de Lyon within the Repid project, and by the EC within the H2020 Program under grant agreement 825333 (MOSAICrOWN).

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
