# OpenReview forum: "WasmTree: Web Assembly for the Semantic Web"
_eswc-conferences.org/ESWC/2021/Conference/Resources_Track — ESWC 2021 Resources_

### Official Review · AnonReviewer2 · 2020-12-29
**WasmTree: Web Assembly for the Semantic Web**

**Rating:** 2
**Confidence:** 2

**Review:**

The paper "WasmTree: Web Assembly for the Semantic Web" studies how the programming language Rust and the recent W3C initiative Web Assembly can contribute to implement efficient Semantic Web libraries for the Javascript ecosystem. The authors propose "WasmTree", a new implementation of the RDF.JS "Store" and "Dataset" interfaces in which the processing is split between the Web Assembly and Javascript layers. In several experiments the authors show that their best setup outperforms state-of-the art implementations for fine-grained data access and SPARQL queries.

All in all, this reviewer considers the paper a solid contribution to the ESWC conference and recommends it for publication.

Update: I read the rebuttals. My score/evaluation remains unchanged.

**Anonymity:**

Yes, I would like my review to remain anonymous.

**Strong Points:**

Interesting topic, very relevant for the ESWC community.

Fresh and novel approach.

Very solid and convincing evaluation.

Very well and clearly written.

**Subreviewer:**

I submitted this review.

**Weak Points:**

If anything, the topic of the paper is perhaps a bit too specific but that is, obviously, not really a reason not to recommend it for publication.

A few language related aspects should be fixed:

- "when high performances are needed" => "when high performance is needed" (the singular, "performance", should be used  in various other places as well)
- "Indexation" => "Indexing"
- "withing" => "within"

In the first sentence of Section 3.1, shouldn't dataset be \texttt{Dataset}?

Please consider putting the three Figures (1, 2, 3) onto the same page to illustrate the differences of the various approaches more clearly.

---

> ### Author Rebuttal · Authors · 2021-01-29
>
> We first thank the reviewer for their feedback on our work.
>
> We are sorry for the remaining typos and will correct them. And we will adjust the position of figures.

---

### Official Review · AnonReviewer3 · 2021-01-11
**Performant in-memory triple store for WASM with clear experiments, but misses some background information**

**Rating:** 2
**Confidence:** 5

**Review:**

### Overview

This resources paper (reusable research prototype) introduces three different implementations of in-memory triple stores using WASM,
with WasmTree being the most performant approach that the authors mostly focus on.
The implementations were done in Rust, and compiled to WASM so that they could be used in a Web-context using JavaScript via the RDF/JS APIs.
The authors show that the WASM implementation can vastly outperform JS implementations, especially regarding memory usage.
This is done using experiments on data ingestion, measuring executing times of separate triple patterns, and measuring execution times of full SPARQL queries.
These last experiments requires the store to be plugged into an existing SPARQL engine, which has been done in both JS and Rust.

Regarding the resources criteria as defined in the CFP, all most relevant criteria appear to be met.
The only one I could not confirm was the sustainability/maintenance plan.

This work certainly contains novel aspects, especially regarding the handling of the WASM-JS connection,
whereby dictionary encoding is handled on the JS-side, and data lookup is handled on the WASM-side.
This approach may in general prove valuable for other WASM implementations.
Furthermore, the paper is clear and well written.
I only have two concerns, which are explained before.
After that, I list several smaller questions and suggestions that may further strengthen this work.

### Concerns

The related work section is well understandable and mostly complete, except for the indexation part.
While HDT is mentioned, I consider it important that more background information is given to other important indexing approaches,
such as RDF-3X, Hexastore, K2-Triples, ...
Furthermore, details should be given on their indexing approaches,
if they persist to disk or are fully in memory,
and how it is equal or different to WasmTree.
Related to this, details on the indexing strategies of Graphy and N3 are missing,
which are crucial for understanding the performance differences in the evaluation section.

In the performance results, it is shown that Oxigraph is shown that 20 times faster than Comunica with WasmTree.
Next it is said that "the efficiency of WASM for complex processing largely pays off compared to the cost of transferring the results back to JS code".
Based on the current data, this is however not necessarily true.
In order to confirm this claim, the authors would need to be certain that Oxigraph and Comunica perform the same operations using the same query plan.
This is because different query plans may lead to very different execution times.
As such, it is possible that the performance difference is caused by the difference in query planning.
While I expect that SPARQL query execution will on average be faster in WASM than JS,
the 20-times-faster result can not be used as an indicator for the claim that was made by the authors.

### Questions and suggestions

In the GitHub README of the SPARQL benchmarks, the following is mentioned:
"the format sent by Oxigraph is not understood by BSBM so the received number of quads is wrong. But we can see on OXigraph's console that quads are actually retrieved, and as we are more interested by speed than developping a proper SPARQL end point, we considered it was good enough for benchmarking purposes."
In ensure result correctness (and the validity of the performance results),
did the authors manually check if the received quads from Oxigraph are in fact correct for all benchmark queries?

On page 4, it is mentioned that 16 sort orders are possible, but only 6 are needed to optimally answer any pattern.
It is important to mention here that this is only true if ordering is not important.
Because there exist join algorithms that assume a certain ordering of results,
which is why it may be beneficial for stores to provide additional indexes.

In the GitHub README of the SPARQL benchmarks, it is mentioned that "This is section is in work in progress".
While the required steps are clear to me with a background in JS and scripting,
this may not be the case for people with different backgrounds.
As such, it would be good to complete its description.

The ingestion experiment is done with 1 million quads.
It may be valuable to also test with different dataset sizes.
This would for example show what the dataset size limits are for the different approaches,
which is something we don't know yet based on the current results.

The evaluation of simple pattern matching only contains patterns for S??? and ?POG.
Many other combinations of patterns exist, which could give other valuable results.
Since this paper also contains performance evaluations for full SPARQL queries,
this is however less important.

"Using the Store interface of WasmTree makes the iteration over all quads 2.73 to 8.22 times slower than using its Dataset interface; it means that in practice, consuming an asynchronous stream of quads is more time consuming than populating a intermediate dataset."
Given the advantages of streaming processing within query engines,
would WasmTree be extendable to expose chunked/buffered streams to offer better performance?

Given the current in-memory storage approach,
would it be possible to (de)serialize the store to disk?

### After Rebuttal

I thank the authors for their rebuttal.

**Anonymity:**

No, I would like my review to be deanonymized.

**Strong Points:**

1. (One of) the first investigations into WASM-JS for RDF.
2. Open-source and reusable implementation of a very performant in-memory triple store.
3. Detailed (and reusable) experiments that measure the performance of this store.

**Subreviewer:**

I submitted this review.

**Weak Points:**

1. Background on RDF indexing/storage could be stronger.
2. Some conclusions and claims are not fully correct.

---

> ### Author Rebuttal · Authors · 2021-01-29
>
> Thank you very much for your review and all your very relevant comments.
>
> The only question we can answer in the rebuttal phase is the manual check of Oxigraph results: yes we did check the results manually, this is the main reason why the quads are printed on the console.

---

### Official Review · AnonReviewer4 · 2021-01-13
**Work with potential but in preliminary state**

**Rating:** 1
**Confidence:** 4

**Review:**

The paper describes the implementation of WasmTree (title), a RUST library to process RDF data according to the RDF.JS specification. The authors motivate such resources by the lack of a library for client-side efficient RDF processing.
The choice of RUST is due to its efficiency and the native cross-compilation into WASM.

To this extent, the authors present three different library implementations that manage the load difference between Javascript and RUST/WASM runtimes. Unsurprisingly, their performance comparison shows that the hybrid approach, which carefully selects which parts of the load executes in JS and which in WASM, outperforms the others.

Although I understand the value of these kinds of work, especially for a resource track, the paper still looks very immature.

### Strong Points
- WASM is a timely technology that might benefit Semantic Web
- The technical quality of the implementation seems high

### Weak Points
- it is written in the form of a technical report rather than a scientific paper
- it related works is very weak and de-focused w.r.t. the actual contribution
- the performance evaluation is very preliminary and, IMHO, not very insightful
- Motivation about performance should be sustained with data
-
In practice, it is almost impossible to assess the potential impact of the resource.

Additionally, the paper does not seem to checks all the guidelines for resources on availability, i.e.,

-   Is the resource (and related results) published at a persistent URI (PURL, DOI, w3id)?
-   Is the resource publicly findable?
-   Is there a plan for the maintenance of the resource?
-   Does it use open standards, when applicable, or have a good reason not to?

## In Depths Comments

### Introduction

This section is very short and does not provide sufficient motivation for the work.

### State of the Art

This section provides a minimal summary of the related technologies. However, it does it in the form of an abstract rather than to give the background information required to understand the paper.

The paper contribution is WasmTree, which relies on specific knowledge about data structures, RDF.JS interfaces, RUST, and WASM.
While it is ok to assume the reader has a certain CS knowledge, too little details are provided about the selected technologies.

This section would have been the right one to present the "WASM principles" on which the resource builds.

### Section 3

This section presents the paper's main contributions. However, the authors present it mixing design decisions, implementation details, and performance insights.

For instance, page 5, last-but-one paragraph:

"We tried to send 1,000,000 integers from WASM to JS: sending in bulk, the whole array was 8 times faster than sending the integers one by one."

This insight should go in the evaluation section, while the authors could elaborate on the alternative models for data processing and formulate hypotheses on their performance. Then, using a proper experiment that evaluates such hypothesis.

It is clear that the technical challenge was carried on with competence. The authors give out details about their decisions in a very honest way. Nevertheless, this approach is not the best for understanding the contribution.

Design, Implementation, and Evaluation should be structured so that it is easier to follow the thought process and understand the premises that made the authors go in a certain way. Otherwise, it becomes a matter of taste to decide if a certain implementation is good or not.

Additionally, I would have expected an algorithmic analysis associated with the work.
Instead, the discussion about how data are processed is done in-line, quite informally, and without sufficient references.

For instance, on page 4, the second bullet point says:

" While up to 16 different sort orders are possible (SPOG, OSPG, OPSG . . . ), 6 sort orders are sufficient to optimally answer any pattern."

Where does this insight come from? A reference or a detailed explanation is missing.

Section 3.4, second paragraph on page 7, second bullet point:

" An identifier list representing the quads in the dataset. Elements of this list are read four at a time (subject, predicate, object, graph name). The role of  this list is to serve as a cache to limit exchanges with the WASM memory."

I have a technical doubt on this part; how do you know what quad are you reading? Are you performing a linear search in the list every time?

### Evaluation

This section presents a quite comprehensive evaluation of the three approaches.
However, it fails to assess the authors' claim: that client-side RDF processing is the bottleneck.

From the experiments, it seems that Graphy, the only external solution, si somehow close to the performance of WasmTree.

Therefore, I am not convinced about the evaluation message without a clear understanding of the design advantages.
Being this is a resource paper, the evaluation should give an idea of the resource's potential impact. Therefore, I encourage the authors to keep working in this direction.


# AFTER REBUTTAL

I'm afraid I have to disagree that WASM was designed to be faster than JS.

    WASI design principles (https://github.com/WebAssembly/WASI/blob/main/docs/DesignPrinciples.md)
    https://github.com/WebAssembly/design/blob/master/Rationale.md

However, I reconsider my comment. The execution speed is probably due to the chosen runtime. RUST is probably the most performant language in this context.

What is missing is a DOI and a citable source for the code. Please add them, here's a guide: https://guides.github.com/activities/citable-code/

Documentation can be improved, but this is a minor issue.

    This is a very small experiment, that is not really related to the Semantic Web: to the end-user of the resource, this result is irrelevant. This sentence, while needed, is actually out of place in every section. It’s more of an explanation about why we go from the first implementation to the second one. We could have said “Web Assembly is better when data is sent in bulk.” but then, readers would have asked for references or a proof. Hence we thought its best place was as a transition between the first and the second solution.

I understand the rationale. Still, this should be fixed somehow. Why don't you rephrase as you suggested and link in a footnote such experiment from the repository? As you said, this experiment is small and complementary.

    The cost to manipulate our structure is directly tied to the costs of manipulation of these structures, so we didn’t mention them to save space.

True. However, javascript is known to underperform in some traditional implementation. Having a study to contrast with the various implementations would help understand what is going on, especially for the hybrid solution. IMHO, the algorithm would still be a relevant contribution for a resource paper, as it makes the work agnostic from the current implementation.

    The identifier list is used as a cache only for operations on which we require all quads from the dataset: iterating on quads, knowing the list of quads. For all other operations, the B-Trees are used instead.

Ok, this clarifies, However, How do you know how to choose the proper data structure? I might have missed this point in the paper.

    The main goal is to build an efficient RDF.JS implementation to achieve better performances than pure-JS implementations.

Agreed, this should better be stated. As I said before, providing the current presentation does not help to understand this claim. In practice, two of the three implementations are baselines for the actual paper contribution.

In conclusion, I changed my mind, and I  raise my grade. However, I encourage the authors to improve the current structure as they highlighted in their comments.



**Anonymity:**

Yes, I would like my review to remain anonymous.

**Strong Points:**


### Strong Points
- WASM is a timely technology that might benefit Semantic Web
- The technical quality of the implementation seems high


**Subreviewer:**

I submitted this review.

**Weak Points:**


### Weak Points
- it is written in the form of a technical report rather than a scientific paper
- it related works is very weak and de-focused w.r.t. the actual contribution
- the performance evaluation is very preliminary and, IMHO, not very insightful
- Motivation about performance should be sustained with data

---

> ### Author Rebuttal · Authors · 2021-01-29
>
> We first thank the reviewer for their feedback on our work.
>
> > Unsurprisingly, their performance comparison shows that the hybrid approach, which carefully selects which parts of the load executes in JS and which in WASM, outperforms the others.
>
> It was actually surprising to us: not only does the hybrid approach outperform every other approach, but a full JS implementation outperforms a full WASM implementation. It means that WASM, which is designed to be faster than JS, is slower than Javascript in our use case.
>
> > Additionally, the paper does not seem to checks all the guidelines for resources on availability,
>
> - We have provided in the article an NPM link to the resource. From it, one can find the github repository. We didn’t include both to save space. We can add the github url in the paper if deemed necessary.
> - In our opinion, the resource is easily findable in NPM and github. Moreover, Google is able to find the resource with the keywords “Wasm Tree RDF”. We are also on the front page with only the keywords “WasmTree” and “Wasm Tree”.
> - We are not sure to understand the open standard concern. To the best of our knowledge RDF, RDF.JS, Web Assembly, Javascript and Rust are open standards / specifications. We use wasm-pack, wasm-bindgen and Graphy which are released under an open source licence, as is our resource.
>
> > This section presents the paper's main contributions. However, the authors present it mixing design decisions, implementation details, and performance insights.
>
> We are presenting three different solutions, two that didn’t work but are the straightforward way to implement the RDF.JS API, and one that worked but with a less straightforward design. We thought performance insights are needed to avoid presenting 3 solutions without the reader questioning why we didn’t stop at the first step.
>
> > "We tried to send 1,000,000 integers from WASM to JS: sending in bulk, the whole array was 8 times faster than sending the integers one by one."
>
> > This insight should go in the evaluation section, while the authors could elaborate on the alternative models for data processing and formulate hypotheses on their performance. Then, using a proper experiment that evaluates such hypothesis.
>
> This is a very small experiment, that is not really related to the Semantic Web: to the end user of the resource, this result is irrelevant. This sentence, while needed, is actually out of place in every section. It’s more of an explanation about why we go from the first implementation to the second one. We could have said “Web Assembly is better when data is sent in bulk.” but then, readers would have asked for references or a proof. Hence we thought its best place was as a transition between the first and the second solution.
>
>
> > Additionally, I would have expected an algorithmic analysis associated with the work. > Instead, the discussion about how data are processed is done in-line, quite informally, and without sufficient references.
>
> We did not provide an algorithmic analysis because we use basic data structures: mainly HashMaps (O(1) operations) and BTrees (Binary trees structures have O(log n) operations, and BTrees is just an optimization to better use the modern processors capabilities). The cost to manipulate our structure is directly tied to the costs of manipulation of these structures, so we didn’t mention them to save space.
>
>
> > " An identifier list representing the quads in the dataset. Elements of this list are read four at a time (subject, predicate, object, graph name). The role of this list is to serve as a cache to limit exchanges with the WASM memory."
> > I have a technical doubt on this part; how do you know what quad are you reading? Are you performing a linear search in the list every time?
>
> The identifier list is used as a cache only for operations on which we require all quads from the dataset : iterating on quads, knowing the list of quads. For all other operations, the B-Trees are used instead.
>
> > From the experiments, it seems that Graphy, the only external solution, is somehow close to the performance of WasmTree.
>
> We also compare our system to N3.JS and Oxigraph. For Simple Pattern Matching, the comparison with N3.JS had to use a different setup than the one with Graphy. Because of space limitations, we could not include the charts for those, but described the results in the text. We can mention in the paper their presence on the Github repo.
> Dataset initialization is fully compared with both N3 and Graphy. Oxigraph is compared in the SPARQL evaluation.
>
> On Fig. 5b WasmTree is one order of magnitude faster than Graphy.
>
> > However, it fails to assess the authors' claim: that client-side RDF processing is the bottleneck.
>
> The main goal is to build an efficient RDF.JS implementation to achieve better performances than pure-JS implementations.

---

> > ### Comment · AnonReviewer4 · 2021-02-11
> > **after rebuttal**
> >
> > I'm afraid I have to disagree that WASM was designed to be faster than JS.
> > - WASI design principles (https://github.com/WebAssembly/WASI/blob/main/docs/DesignPrinciples.md)
> > - https://github.com/WebAssembly/design/blob/master/Rationale.md
> >
> > However, I reconsider my comment. The execution speed is probably due to the chosen runtime. RUST is probably the most performant language in this context.
> >
> > What is missing is a DOI and a citable source for the code. Please add them, here's a guide: https://guides.github.com/activities/citable-code/
> >
> > Documentation can be improved, but this is a minor issue.
> >
> > >This is a very small experiment, that is not really related to the Semantic Web: to the end-user of the resource, this result is irrelevant. This sentence, while needed, is actually out of place in every section. It’s more of an explanation about why we go from the first implementation to the second one. We could have said “Web Assembly is better when data is sent in bulk.” but then, readers would have asked for references or a proof. Hence we thought its best place was as a transition between the first and the second solution.
> >
> > I understand the rationale. Still, this should be fixed somehow. Why don't you rephrase as you suggested and link in a footnote such experiment from the repository? As you said, this experiment is small and complementary.
> >
> > > The cost to manipulate our structure is directly tied to the costs of manipulation of these structures, so we didn’t mention them to save space.
> >
> > True. However, javascript is known to underperform in some traditional implementation. Having a study to contrast with the various implementations would help understand what is going on, especially for the hybrid solution. IMHO, the algorithm would still be a relevant contribution for a resource paper, as it makes the work agnostic from the current implementation.
> >
> > > The identifier list is used as a cache only for operations on which we require all quads from the dataset: iterating on quads, knowing the list of quads. For all other operations, the B-Trees are used instead.
> >
> > Ok, this clarifies, However, How do you know how to choose the proper data structure? I might have missed this point in the paper.
> >
> > > The main goal is to build an efficient RDF.JS implementation to achieve better performances than pure-JS implementations.
> >
> > Agreed, this should better be stated. As I said before, providing the current presentation does not help to understand this claim. In practice, two of the three implementations are baselines for the actual paper contribution.
> >
> > In conclusion, I changed my mind, and I will raise my grade to be borderline. However, I encourage the authors to improve the current structure as they highlighted in their comments.

---

### Official Review · AnonReviewer1 · 2021-01-15
**RDF meets WASM (finally)**

**Rating:** 2
**Confidence:** 5

**Review:**

# Summary
The paper describes an original approach to leveraging WebAssembly to reduce storage overhead and improve retrieval performance of RDF data from a JavaScript environment such as Node.js or the Web browser.

# Overall Evaluation
The paper is clearly written and addresses virtually all the relevant points and covers all the right details I would expect to find from such a study. This is much easier to do when the scope is narrow, which the authors have effectively used to their advantage. The results of the experiments are also undoubtedly valuable to the community.

## Comments to the authors
Several years ago, I too ventured down the WASM path to see if I could improve graphy's performance but gave up after experimentation revealed the tremendous penalty that must be payed for transcoding data and reconstructing objects in memory between the JavaScript runtime and WASM (including the UTF-8 <--> UTF-16 transcoding). This lead me to believe that WASM performance gains were much more difficult to take advantage of considering that users of Web libraries will always be authoring their application logic in the more forgiving JavaScript, and that WASM would be there for some CPU-intensive tasks on large enough datasets that would make it worth the penalty. I am pleased to see this work find a cooperative data structure that is able to take performance gains from WASM. The other thought I had from my experiments is that one day, there will be many more Web APIs accessible to WASM and the whole web application can run in WASM. At such a point, we might expect JavaScript RDF libraries to become obsolete.

## Minor Points

### Using 6 Indexes
I am amazed to still see projects using 6 indexes for triplestore implementations. The authors of HDT have demonstrated time and again that using 6 indexes offers barely any performance benefits (if any at all in practice) and at more than twice the storage cost compared to HDT's unique 3-index wavelet data structure. I have yet to see someone prove me wrong about this. I bring it up for posterity, but maybe the authors could also incorporate this in the future work section.

### Future Work
Have the authors considered the potential of leverage multiple threads? Would be very interesting to see how far we can push performance in the browser with [multithreaded parsers](https://github.com/blake-regalia/graphy.js/blob/master/perf/README.md#test_count_nt) and WASM boosts.

### Evaluation
- "WasmTree is a bit slower than both N3.js and Graphy" <-- more than "a bit" for graphy :) You have space on this page, why not simply quantify the differences? E.g., A is 0.8x as fast as B, etc.
- "In real world applications, the lazy indexing strategy will ensure the users benefit from the fastest initialization time, amortizing the cost of creating the missing indexes over the use of the dataset." <-- this is a very important point (with unfortunately limited supporting evidence) that should resurface in the conclusion.
- "a B-Tree is equivalent to a sorted array: so it only saves the cost of a single exchange between JS and WASM. " <-- I must admit i don't understand this statement
- "being an easier-to-use library, as users do not have to care about freeing memory except for the datasets"  <-- I would generally try to avoid subjective terms like "easier-to-use" in an academic paper without user evaluation

## Typos

### Section 2 / Web Assembly
 - "allocated by JS code as an array of integers" -- partially true; the ArrayBuffer is an array of *bytes*, the various DataViews provide interfaces to read that buffer as integers but most runtimes will optimize away ES number bloating and use pure `uint8`, `uint16, or `uint32` for certain subsequent operations on the ArrayBuffer.

### Section 3.1 / last paragraphs
- "WASM memory to the JS code, is ~declined~ in three..." <-- defined? described?
- "The naive TreeDataset approach ~consists in~ a full-Rust..." <-- incorrect grammar here

### Section 4.3
 - There are some comma (,) decimal separators used in Table 1
 - I find the sparse matrix to be a somewhat unintuitive way to present this information for such a small set of comparisons, but that could just be me...
 - "because it only uses WASM memory for BTrees, and we saw in Section 4.1 that those only
represent a small fraction of the memory needed to store a dataset" <-- we did? What exactly is that fraction? Be explicit, restate if necessary.

# Response to Authors' rebuttal
> If by “unique 3 indexes”, you mean that HDT uses 3 trees, it is actually similar to what we are doing. HDT stores graphs while we store datasets, with the extra Graph component to the triple. HDT keeps 3 of the possible 6 orderings of SPO; we keep 6 of the possible 24 orderings of SPOG In both case, all possible patterns can then be efficiently queried.

I suggest taking a closer look at the more technical HDT publications, especially those which detail wavelet trees and the HDTQ paper. The traditional way of thinking about indexes does not apply to RDF in practice, and HDT(Q) has been designed to exploit typical RDF graph shapes to achieve virtually the same performance as the old "Hexastore" approaches without the combinatorial number of indexes needed. In short, 6 indexes is most likely always overkill in practice, and people continue to miss this very important point.

> When there are few elements, the identifier list...

My point about the excerpt is that it is not at all clearly stated in the paper. Spell out exactly what you are trying to convey in the paper, not only to me in the rebuttal  :)

> should indeed be restated into “As hinted by section 4.1”.

Or not at all? Consider the value it adds if any.

> because the memory measure is imperfect

You should still be able to measure and report something; how about using a margin of error?



**Anonymity:**

No, I would like my review to be deanonymized.

**Subreviewer:**

I submitted this review.

---

> ### Author Rebuttal · Authors · 2021-01-29
>
> We first thank the reviewer for his feedback on our work.
>
> > I am amazed to still see projects using 6 indexes for triplestore implementations. The authors of HDT have demonstrated time and again that using 6 indexes offers barely any performance benefits (if any at all in practice) and at more than twice the storage cost compared to HDT's unique 3-index wavelet data structure.
>
> About the 6 indexes parts, we are not sure about what you mean about HDT.
> If by “unique 3 indexes” you mean that there is only one tree, HDT seems to use several indexes with one main index and at least one O?? index (https://www.rdfhdt.org/hdt-internals/ section 5)
>
> If by “unique 3 indexes”, you mean that HDT uses 3 trees, it is actually similar to what we are doing. HDT stores graphs while we store datasets, with the extra Graph component to the triple. HDT keeps 3 of the possible 6 orderings of SPO; we keep 6 of the possible 24 orderings of SPOG In both case, all possible patterns can then be efficiently queried.
>
> > *"Is it not really surprising as for **small datasets**, up to 2B − 1 elements to be precise, * "a B-Tree is equivalent to a sorted array: so it only saves the cost of a single exchange between JS and WASM. " <-- I must admit i don't understand this statement
>
> When there are few elements, the identifier list representation in Javascript and the BTree representation are basically the same representation : an array of 28 integers. It is the case when the match function is called with a S??? pattern on the Person Dataset: 7 quads are returned in one array.
>
> In the default WasmTree implementation, the identifier list is already in the JS environment, so it does not call a WASM function to retrieve it.
>
> The “WasmTree Always Forest” implementation retrieves the identifier list from the WASM environment. When there are less than 2B elements, it consists in copying the tree root element. This is the only difference between the two, and explains the small measured difference in time, as shown in figure 6a.
>
> We can precise this in the paper.
>
> > "because it only uses WASM memory for BTrees, and we saw in Section 4.1 that those only represent a small fraction of the memory needed to store a dataset" <-- we did? What exactly is that fraction? Be explicit, restate if necessary.
>
> “We saw in section 4.1” should indeed be restated into “As hinted by section 4.1”.
> We can’t explicitly quantify the fraction because the memory measure is imperfect, as stated in the last paragraph of that section. We can deduce it from Figure 4a with the difference of memory used by TreeDataset 1 index / TreeDataset 6 indexes (which are pure Web Assembly implementations) and WasmTree 1 index / 6 indexes.

---

> > ### Comment · AnonReviewer1 · 2021-02-13
> > **Response to Rebuttal**
> >
> > > If by “unique 3 indexes”, you mean that HDT uses 3 trees, it is actually similar to what we are doing. HDT stores graphs while we store datasets, with the extra Graph component to the triple. HDT keeps 3 of the possible 6 orderings of SPO; we keep 6 of the possible 24 orderings of SPOG In both case, all possible patterns can then be efficiently queried.
> >
> > I suggest taking a closer look at the more technical HDT publications, especially those which detail wavelet trees and the HDTQ paper. The traditional way of thinking about indexes does not apply to RDF in practice, and HDT(Q) has been designed to exploit typical RDF graph shapes to achieve virtually the same performance as the old "Hexastore" approaches without the combinatorial number of indexes needed. In short, 6 indexes is most likely always overkill in practice, and people continue to miss this very important point.
> >
> > > When there are few elements, the identifier list...
> >
> > My point about the excerpt is that it is not at all clearly stated in the paper. Spell out exactly what you are trying to convey in the paper, not only to me in the rebuttal  :)
> >
> > > should indeed be restated into “As hinted by section 4.1”.
> >
> > Or not at all? Consider the value it adds if any.
> >
> > > because the memory measure is imperfect
> >
> > You should still be able to measure and report something; how about using a margin of error?

---

### Official Review · AnonReviewer5 · 2021-01-16
**The proposed framework has promising performance but the paper needs a major revision.**

**Confidence:** 5

**Review:**

The paper presents WasmTree a framework for manipulating RDF data at both client and server side. The framework aims at streamlining performance in storing, manipulating and retrieving RDF data in the javascript ecosystem. The name of the framework refers (I suppose) to the fact that quads are stored using B-Trees (a data structure usually adopted for indexing database entries). The way quads are actually indexed is described in Section 3. The authors present three performance evaluations: 1) time and memory needed for loading a dataset 2) time and memory needed for evaluating simple graph patterns (e.g. “find all resources of type Person”); 3) time and memory needed for evaluating SPARQL queries from the Berlin SPARQL benchmark.

The problem of providing efficient frameworks for manipulating RDF data in a javascript runtime environment is clearly important. Adopting efficient technologies (such as, RUST and WebAssembly) is a direction, which is worthwhile further investigations. The proposed framework goes in this direction, but the paper needs a major revision, before it can fulfill its purpose.

Concerning clarity, I would suggest the authors to rework the introduction in order to clarify what are the context, motivation and goal of the work. Why is this framework needed? What gaps does this framework fills? Why are existing frameworks insufficient? I read that this work has been developed within the context of a research project. This project could provide the authors with useful use cases for motivating their work.

Concerning the state-of-the-art section, I would say that this is not the state-of-the-art of the problem at hand, but a sort of background of the work. The authors mainly list a series of definitions or descriptions of frameworks with no discussion of the connections with the proposed framework. Instead, a qualitative comparison with the existing frameworks is needed.

Concerning the approach, it was not clear to me if B-Trees are pre-computed or not. If the B-Trees are pre-computed, what happens when a new triple is added/removed/updated. How much do these operations cost? If the B-Trees are not pre-computed, when do the trees are computed?

Considering significance, first of all I do appreciate the work invested by the authors in devising their approaches as well as implementing and evaluating them. However, the evaluations presents weak points. As for the first evaluation, the authors selected 1M quads from “the Person data dataset of the 2016-10 dump of DBpedia”. Why did you select only the Person data? Evaluating the performance against a random sample from the LOD (not only DBpedia) would have been more appropriate. As for the second evaluation, what competency questions did you use? Why these queries and not others?

Review by Luigi Asprino and Valentina Presutti

**Anonymity:**

No, I would like my review to be deanonymized.

**Rating:**

-2: Reject

**Strong Points:**

importance of the topic
interesting direction of research

**Subreviewer:**

I delegated this review to a subreviewer.

**Weak Points:**

The paper needs a major revision.
The clarity of the paper can be improved.
The authors don’t provide a proper motivation for the work.
The paper misses a proper discussion of the related work.
There are some flaws in the evaluation method.

---

> ### Author Rebuttal · Authors · 2021-01-29
>
> We first thank the reviewers for their feedback on our work.
>
> > Concerning the approach, it was not clear to me if B-Trees are pre-computed or not. If the B-Trees are pre-computed, what happens when a new triple is added/removed/updated
>
> Only one B-Tree exists at first, and other B-Trees are lazily created when the user searches for a corresponding pattern. We are following the RDF.JS Dataset interface: at any time, the user can add or remove quads. The B-Trees that exist at that time are updated.
>
> > How much do these operations cost?
>
> The cost of an operation is the cost of an operation in the described structures. For example, the mean complexity of insertion in a BTree is log n, the complexity of insertion in a HashMap is O(1).
>
> > As for the second evaluation, what competency questions did you use? Why these queries and not others?
>
> We are not sure to understand what you mean by “competency questions”. The main goal is to build an efficient RDF.JS implementation to achieve better performances than pure-JS implementations.
>
>
> The second evaluation with Simple Pattern Matching requests some arbitrarily chosen  patterns, which is a basic operation permitted by the RDF.JS interface.
>
> The third evaluation uses SPARQL select queries to test multiple patterns (including those not tested in the previous evaluation) on a more diverse dataset by using the BSBM Explore Use Case. This evaluation lets us benchmark the resource in a more realistic context: as a component of a bigger library.

---

### Decision · Program_Chairs · 2021-02-23

**Decision:**

Accept

**Comment:**

The overall judgment about this paper is good and there is a certain agreement by the Reviewers for accepting this paper.
The authors are anyway asked for carefully following the indications provided by each Reviewer in order to improve the final version of the manuscript.